# Determination of Fatty Acids Profile in Original Brown Cows Dairy Products and Relationship with Alpine Pasture Farming System

**DOI:** 10.3390/ani10071231

**Published:** 2020-07-20

**Authors:** Stella Agradi, Giulio Curone, Daniele Negroni, Daniele Vigo, Gabriele Brecchia, Valerio Bronzo, Sara Panseri, Luca Maria Chiesa, Tanja Peric, Doina Danes, Laura Menchetti

**Affiliations:** 1Department of Veterinary Medicine, University of Milan, Via dell’Università 6, 26900 Lodi, Italy; stella.agradi@studenti.unimi.it (S.A.); negroni.vet@gmail.com (D.N.); daniele.vigo@unimi.it (D.V.); gabriele.brecchia@unimi.it (G.B.); valerio.bronzo@unimi.it (V.B.); 2Department of Health, Animal Science and Food Safety “Carlo Cantoni”, University of Milan, Via Celoria 10, 20133 Milan, Italy; sara.panseri@unimi.it (S.P.); luca.chiesa@unimi.it (L.M.C.); 3DI4A—Dipartimento di Scienze Agroalimentari Ambientali e Animali/Department of Agricultural, Food, Environmental and Animal Sciences, University of Udine, Via Sondrio 2/B, 33100 Udine, Italy; tanja.peric@uniud.it; 4Facultaty of Veterinary Medicine, University of Agronomic Sciences and Veterinary Medicine, Splaiul Independentei 105, 050097 Bucharest, Romania; danes.doina@gmail.com; 5Department of Veterinary Medicine, University of Perugia, Via San Costanzo 4, 06126 Perugia, Italy; laura.menchetti7@gmail.com

**Keywords:** Original Brown cow, milk, cheese, grazing, alpine pasture, fatty acids, lipids

## Abstract

**Simple Summary:**

In the world context of climate change and increasing food needs, it is of vital importance to preserve genetic resources in production ecosystems. On the Italian Alps that means conservation of local cattle breeds and maintaining of traditional alpine agro-pastoral practices. In particular, positive effects, such as preservation of fauna and flora biodiversity and greater appreciation by tourists, have been shown to correlate with mountain pastures. This study aimed to evaluate the relationships between different fatty acids in Original Brown cows’ milk and cheese as well as the pattern of fatty acids that most contributes to discriminate between two different farming systems, in which the main difference was the practice, or not, of alpine summer-grazing. A multivariate statistical approach was used. Results indicated that the diet fed to animals, by influencing ruminal fermentation process and mammary gland de novo synthesis, is responsible for defining fatty acids characteristic profiles in dairy products. Findings also suggest that it is possible to build a reliable model based on specific fatty acids concentrations in order to identify pasture-raised or not dairy products. By this way, farmers who practice traditional agro-pastoral system could be easily recognized and adequately remunerated.

**Abstract:**

This study aimed to evaluate the relationships between fatty acids and the pattern that most contributes to discriminate between two farming systems, in which the main difference was the practice, or not, of alpine summer-grazing. Milk and cheese were sampled every month in two farms of Original Brown cows identical under geographical location and management during no grazing season point of view in the 2018 season. Fatty acids concentrations were determined by gas chromatography. The principal component analysis extracted three components (PCs). Mammary gland de novo synthetized fatty acids (C14:0, C14:1 n9, and C16:0) and saturated and monosaturated C18 fatty acids (C18:0, C18:1 n9c) were inversely associated in the PC1; PC2 included polyunsaturated C18 fatty acids (C18:2 n6c, C18:3 n3) and C15:0 while conjugated linoleic acid (CLA n9c, n11t) and fatty acids containing 20 or more carbon atoms (C21:0, C20:5 n3) were associated in the PC3. The processes of rumen fermentation and de novo synthesis in mammary gland that are, in turn, influenced by diet, could explain the relationships between fatty acids within each PC. The discriminant analyses showed that the PC2 included the fatty acids profile that best discriminated between the two farming systems, followed by PC3 and, lastly, PC1. This model, if validated, could be an important tool to the dairy industry.

## 1. Introduction

The annual average abandonment rates of farms in the Italian Alps is still growing, according to a recent agricultural census, and the trend is getting worse compared to the two decades before 2000 (from −3.4% to −4.5%) [1]. These agro-structural changes are linked to little or no use of mountain pastures or to their utilisation for mowing rather than for grazing. All this is responsible for the loss of open areas and forest re-growth, that lead to reduction of fauna and flora biodiversity as a result [2,3,4]. Furthermore, keeping the traditional alpine agro-pastoral practices is related to a greater appreciation by tourists and produce positive socio-economic implications, helping the environmental and economic sustainability which is one of the future challenges for the agri-food systems [5,6].

Unfortunately, not only farms are being abandoned, but also local cattle breeds. These ones are usually limited to a small number of animals concentrated in the same geographical area, which typically represents their cradle of origin. In fact, these specific areas are the environment where the native breeds have been selected through decades, if not centuries. Thereby, these local breeds find, in these habitats, the environment to which they best suit and in which they maximise their quantitative-qualitative production potential. The union among bovine biodiversity (different breeds means different milks), territories (different animal feed), and different traditions led to the creation of the thousands of products that characterize the worldwide famous Italian diet. Despite this, in the past decades, the inferior productive capacity of local breeds compared to other cosmopolitan ones has caused their gradually consistent reduction, and, sometimes, extinction [7,8,9,10,11,12]. This trend means loss of single breed genetic heritage, and, also, loss of across-breed genetic variability [13,14,15,16]. However, maintaining genetic diversity is also important considering the actual climate change, both in order to permit the selection of a specific animal that better fit a new environment and to select for robustness, in general, to create animals that can fit a range of environmental conditions [14].

The Original Brown (or Original Braunvieh or Bruna alpina originale) is a native breed present mainly in Switzerland, Austria, and Italy (especially in the provinces of Bolzano and Bergamo) [17]. It differs from the Brown (or Improved Braunvieh or Bruna alpina) because the latter derives from the cross of Original Brown with Brown Swiss, which in turn derives from the Original Brown exported to USA between 1869 and 1910 and there selected in order to increase milk production [18]. Since 2015 the Italian National Breeders’ Association of Brown Breed manages the register of meat line subjects (i.e., Original Brown subjects). In 2017 the living registered animals were 1500 [17], but no studies have been published about dairy products quality of the breed on Italian Alps, in particular considering its relationship with grazing system.

The Original Brown, even though it has a lower average milk yield than the other two breeds, has better performance in terms of fertility, mammary gland health, longevity [19], and a higher genetic diversity, also compared to many other mainstream cattle breeds [18,20]. However, a study has demonstrated that in a low-input system these three breeds have a similar milk yield, probably due to the difficulty of the Brown and Brown Swiss to fully exploit their productive potential. At the same time, between the three breeds, Original Brown was the one with higher n3/n6 ratio and concentrations of several nutritionally desirable fatty acids during grazing season [21]. Thus, on the Alps and in extensive grazing system, Original Brown breeding could be considered an optimal choice in order to produce high quality dairy products [21]. In fact, in particular with reference to major milk fatty acids, scientists have recognised an effective or potential beneficial role to C18:1 n11t [22,23,24,25,26], conjugated linoleic acids (CLA) [27], and to a correct n6/n3 ratio [28].

Different studies have shown that milk produced with grazing systems in the Alps, rather than in lowland systems or with an hay based diet, is characterized by an optimal fatty acids profile from human health point of view [29,30]. Therefore, since the products derived from grazing are particularly rich in the beneficial classes of fatty acids, compared to the ones from many other feeding systems, they have an added value, considering, also, the other advantages of the use of pastures. Moreover, it has been shown that northern Italian consumers recognise the relationship between type of feed and dairy products quality and tend to prefer milk coming from grazed cows.

For these reasons the possibility to identify the feeding system which gives rise to a dairy product could be an important tool to the dairy industry in order to ensure consumers know about the origin. On the other hand, the farmers who practice traditional agro-pastoral system could be easily recognised and adequately remunerated.

Different studies have already demonstrated the possibility of the utilization of fatty acids profile from dairy products not only to identify the type of feeding system (daily rations of fresh grass vs. no fresh grass in the diet) [31,32,33,34,35,36], but also to ensure geographical authentication of the protected designation of origin dairy products [37]. However, despite the breed consistency increasing importance in last years, no studies have considered the possibility to distinguish between dairy products of Original Brown cows grazing or not on Italian alpine pastures.

The aim of this work was to characterize, by using a multivariate approach, the fatty acid profile of milk and cheese from Original Brown cows produced in two farming systems, one that practiced grazing and the other not. First, the relationships between fatty acids in milk and cheese from the two farms were defined by principal component analysis. Then, the pattern of fatty acids in milk and cheese that most contributes to discriminate the two farming systems was identified by discriminant analyses.

## 2. Materials and Methods

### 2.1. Farm Conditions, Animals, and Cheesemaking Procedure

Two farms, placed in the same geographical area, both breeding Original Brown cows managed and fed with identical hay and concentrate (during no-grazing season), with analogous cheesemaking procedure, were selected and studied during the same period in order to limit the variability linked to every factor that was not due to the grazing practice.

The family-run farms are located in the same mountainous municipality (Valgoglio, BG, Italy, about 1000 m.a.s.l.). Both keep on average 15 cows of Original Brown breed in lactation for the production of cheese and meat.

In “A” farm, grazing is practised from May 10th to November 10th every year on a foothill pasture between 900 and 1200 m.a.s.l. During this period the animals are fed green fodder essences and a saline supplementation is available. Concentrate is provided just to individual subjects in case of need. In the other six months of the year the diet consists of local polyphytic hay (18−20 kg/day, 60% first cut, 40% second cut) and concentrate (3.5−5 kg/day; Appendix A), and the cows are housed in a tie-stall barn. Cows in this farm had an average of 289 days in milk during the sampling period. Considering the distribution of parity, there were two peaks, the first between May and April and the second between November and December.

In the “B” farm, the cattle are housed throughout the year in a tie-stall barn, and they receive the same rations of the farm “A” during no-grazing season. From mid-August to the end of September the diet is supplemented by fresh grass of local origin (Figure 1). Cows in this farm had an average of 295 days in milk during the sampling period. The distribution of parity had just a peak along the year between November and December.

Cheesemaking is performed with unpasteurized, full-fat milk. Both farms produce a half-cooked (41−42 °C) semi-hard cheese with the use of animal rennet. The ripening, which is carried out on farms, varies between 20 and 40 days, but 30 days is the ideal duration. During this period, cheeses are kept under the same environmental conditions (temperature and relative humidity) in both farms.

### 2.2. Milk and Cheese Sampling

From January 2018 to December 2018 the bulk milk of both farms was collected on the 1st of every month after the evening milking in Corning™ Falcon 15 mL conical centrifuge tubes. The samples were immediately refrigerated and sent to the laboratory in 12−18 h, where they were frozen and stored at −20 °C until analysis.

The cheese was sampled on the 30th day of ripening (a 100 g slice with both rind and paste) along the same period. The sample was put in a plastic bag and followed the same procedures as milk samples.

### 2.3. Milk and Cheese Samples Analysis

Milk and cheese samples fatty acids composition was determined by gas chromatography (GC) according to Bligh and Dyer method [38] and International Organization for Standardization (ISO) [39] as has been reported previously by Papaloukas et al. [40]. A total of 37 fatty acids (16 saturated fatty acids, 9 monounsaturated fatty acids, 7 n6 polyunsaturated fatty acids, 4 n3 polyunsaturated fatty acids, and 1 stereoisomer of conjugated linoleic acid) were evaluated. Individual fatty acids were used also to obtain the sums of the following categories of fatty acids: saturated fatty acids (SFA), monounsaturated fatty acids (MUFA), and n3 and n6 polyunsaturated fatty acids (n3 and n6 PUFA). The concentrations of fatty acids were expressed as mg/100 mg total fatty acids. All samples were analysed in duplicate.

### 2.4. Statistical Analysis

Initial, descriptive statistics were used to present the fatty acids composition.

Then, principal component analysis (PCA) was used to describe relationships between fatty acids and to reduce the original variables into a smaller number of fatty acid profiles. First, a set of fatty acids were selected based on their biological relevance including C10:0, C12:0, C14:0, C15:0, C16:0, C18:0, C21:0, C14:1 n9, C16:1 n9, C18:1 n9c, C20:1 n9, C18:2 n6c, C20:4 n6, C18:3 n3, C20:5 n3, and CLA n9c, n11t. Then, a correlation matrix was constructed to identify very low or very high correlations and identify suitable variables for the component (PC) (Appendix A) [41,42]. Multicollinearity was further verified via the determinant of the correlation matrix. The sampling adequacy and the sphericity were verified using Kaiser–Meyer–Olkin (KMO) and Bartlett’s tests, respectively. A KMO > 0.6 indicated an adequate factorability. PCs showing eigenvalues greater than 1 were retained and rotated with the Varimax method. Corresponding PC scores were calculated using the regression method to obtain three new variables as linear combinations of the original variables of fatty acids. Factor loadings with an absolute value greater than 0.5 were interpreted and used to assign a PC label [42,43,44].

Then, the effect of farming system on these PCs describing fatty acid profiles was evaluated using generalized linear models (GLMs) stratified by product (milk and cheese). In the GLMs, the month (12 levels: January-December) was included as within-subject effects, the PCs as dependent variables, and the farm (2 levels: Grazing and no grazing) as predictor. Normal and identity were the probability distribution and the link function, respectively.

Moreover, a discriminant analysis (DA) was performed in milk and cheese products to find the combinations of fatty acids that distinguish grazing and no grazing and to assess the relative importance of each fatty acid in classifying the farming system. Indeed, DA found the linear combinations of fatty acid profiles (discriminant function (Df)) that best discriminate the farms. The three PCs were included in DA as independent variables. Mahalanobis distance was used to verify multivariate normality and to identify the presence of multivariate outliers. The relative importance of each variable in classifying the farming system was evaluated by the Wilks’ lambda (the smaller the Wilks’ lambda, the more important the variable to the Df) and by the standardized Df coefficients (analogous to beta coefficients in multiple regression). The centroids, indicating the mean discriminant scores of Df for grazing and no grazing, were used to establish the cutting point for classifying samples. The performances of the final Df were estimated by running a leave-one-out cross-validation, which calculates the probability for each sample to be accurately classified in the correct farming system [45,46].

Finally, we used chi square (χ^2^) goodness of fit test to evaluate whether there was a prevalent category of fatty acids.

Statistical analyses were performed with SPSS Statistics version 25 (IBM, SPSS Inc., Chicago, IL, USA). Statistical significance occurred when *p* < 0.05.

## 3. Results

### 3.1. Milk and Cheese Fatty Acids Composition

In both farms and in both products (milk and cheese) saturated fatty acids were the most representative class of fatty acids, followed by monosaturated fatty acids and n6 polyunsaturated fatty acids, while n3 polyunsaturated fatty acids and the stereoisomers of conjugated linoleic acid were the lowest classes and had similar percentage (χ^2^ = 167.6 and χ^2^ = 170.7 in milk and cheese, respectively; *p* < 0.001). The fatty acids which had higher concentrations where, in order, C16:0, C18:1 n9c, C14:0, and C18:0. In general, the major quantitative fatty acids were the saturated C10:0, C12:0, C14:0, C16:0, and C18:0, the monosaturated C16:1 n9, C18:1 n6c, 18:1 n9c, C18:1 n9t, the polyunsaturated C18:2 n6c, and C18:3 n3, and the stereoisomer of conjugated linoleic acid CLA n9c, n11t. The n6/n3 ratio was less than two and a half, both in milk and cheese (Table 1 and Table 2).

### 3.2. Principal Component Analysis

After inspection of the correlation matrix, the following variables were included in the PCA: C14:0; C14:1 n9; C15:0, C16:0; CLA n9c, c11t; C18:0; C18:1 n9c; C18:2 n6c; C18:3 n3; C20:5 n3, and C21:0 (Appendix A). The PCA extracted three principal components, which together account for 76.4% of the variance (Table 3). The KMO = 0.70 indicated the sampling adequacy of the PCA. Moreover, as showed in Figure 2, the variables appeared well separated in the rotated space.

The PC1, which explained a large proportion of the total variance (45.7%), was a bipolar component. In this PC, saturated (C18:0) and monounsaturated (C18:1 n9c) fatty acids with 18 carbon atoms had high positive loadings and were opposed to the fatty acids with less than 18 carbon atoms (C16:0, C14:0, and C14:1 n9). Thus, we named this bipolar PC as “C < 18 − C18”. The PC2 included n3 and n6 series of PUFA with 18 carbon atoms as well as pentadecanoic acid (C15:0). Finally, in the PC3, the highest loadings were found for fatty acids containing 20 or more carbon atoms in addition to the conjugated linoleic acid. The PC2 and PC3 were called “C18 PUFA & C15” and “C > 18 & CLA”, respectively.

### 3.3. Characterization of Fatty Acid Profile on Milk and Cheese from Grazing and No Grazing Dairy Farming Systems

Both milk and cheese from no grazing farming system had lower PC2 (*p* < 0.001) and PC3 than grazing system (*p* < 0.05; Figure 3). Conversely, PC1 was not affected by the farming system, especially with regards to cheese (*p* = 0.055 and *p* = 0.324 for milk and cheese, respectively; Figure 3).

In the DA, Mahalanobis distance did not identify any multivariate outlier. The standardized coefficients calculated by the DA indicated that PC2 was the dominant variable for the function. A high contribution was also ascribed to the PC3, more significant in milk than cheese (Table 4). This finding suggests that PUFA and C15:0 as well as C > 18 and CLA are the fatty acids that contribute most to characterize the products of the two farming systems. The lowest coefficients (and the higher Wilks’ lambda) were found for PC1 suggesting that saturated and monounsaturated fatty acids with 18 or lower carbon atoms have a minor importance in discriminating the farming system, especially for cheese.

The group centroids confirmed that the two farm systems have a noticeable mean difference on fatty acids profile of milk (1.329 and −1.329 for grazing and no grazing system, respectively) and cheese (1.210 and −1.210 for grazing and no grazing system, respectively; Appendix A). In particular, the differences concern the variables with high Df coefficients, PC2 and PC3. Thus, grazing farm products had high content of PUFA, C15:0, C > 18, and CLA while the no grazing farm products had below average content of these fatty acids.

Distributions of the PC2 and PC3 scores in the factor map supported this result (Figure 4). This graph shows that the products of the no grazing farm are concentrated in the third quadrant in which both coordinates are negative. Moreover, it is worthwhile noting that the scores of the no grazing farm are very close to each other while those of the grazing farm are more scattered suggesting that the latter had higher variability in PC2 and PC3 scores.

After cross-validation, the 83.3% and 87.5% of milk and cheese samples, respectively, were correctly classified in their original farming system. Four samples of milk were misclassified (three for grazing and one for no grazing farm; Table 5). The samples of the grazing farm that were not correctly classified were collected in the months of January, March, and April. The incorrectly classified milk sample from no grazing farm was collected in October.

As regards the cheese, three samples from the grazing were classified incorrectly (Table 5). They were collected during February, April, and November. Therefore, the incorrectly classified samples mainly came from the grazing farm (*n* = 6), confirming a greater variability in their fatty acids composition.

## 4. Discussion

This study evaluated the lipid composition of milk and cheese from Original Brown cows by using a multivariate approach which allowed to (i) describe the association dynamics between different fatty acids and (ii) identify the fatty acids profile that best discriminates the products according to the farm system.

To our knowledge, only Stergiadis et al. [21,47] have previously evaluated the fatty acids composition of milk from Original Brown cows in Switzerland. Our study’s main fatty acids’ qualitative-quantitative composition is consistent with the one of Stergiadis. However, compared to our study, Stergiadis’ ones took place in a different geographical area, did not consider the cheese fat composition, and the ration fed to Original Browns was more variable.

### 4.1. Relationships between Different Fatty Acids in Milk and Cheese: The Principal Component Analysis

The PCA aimed to evaluate the relationships between the different fatty acids, select a short number of fatty acids which characterize the products, and find a small number of underlying dimensions describing them [46]. The PCA extracted three PCs in which fatty acids seem associated according to the processes of rumen fermentation and de novo synthesis in mammary gland that are, in turn, influenced by diet.

The first component was bipolar as C14:0, C14:1 n9, and C16:0 negatively associated with C18:0 and C18:1 n9c. The first three fatty acids derive entirely (C14:0 and C14:1 n9) or mostly (C16:0) from de novo synthesis in mammary gland [48,49]. Instead, C18:0 and C18:1 n9c arise nearly completely from blood lipoproteins, which mostly have intestinal provenance and, in a lesser extent, derive from triacylglycerols of endogenous origin [50]. A recent meta-analysis has confirmed the inhibitory effect of some fatty acids deriving from ruminal biohydrogenation of C18 fatty acids towards milk fat de novo synthesis [51]. A cattle diet based on forages, that have a high content of C18 fatty acids [52], is responsible for an increase in milk concentration of both C18:0 and C18:1 n9c. Indeed, they follow the same trend as C18:0, the final product of the C18 fatty acids biohydrogenation reactions in the rumen [53], is partially Δ9-desaturated in the mammary gland and adipose tissue [54]. Simultaneously this diet leads to an increase of the ruminal biohydrogenation reaction intermediates of C18 fatty acids and, therefore, to a decrease in the mammary gland synthesis of C14:0, C14:1 n9, and C16:0. This could explain the “bipolar behaviour” of C18 fatty acids and de novo synthetized fatty acids (C14:0, C14:1 n9, and C16:0) in the PC1.

The second dimension was constituted by C18 polyunsaturated fatty acids (C18:2 n6c and C18:3 n3) and C15:0. The first two fatty acids cannot be synthetized by the mammals [55]. When reaching the rumen microbiota, these PUFAs undergo a biohydrogenation process which leads to the production of a wide range of different reaction intermediates. The C18:3 n3 and C18:2 n6c found in milk derive from the portion that bypasses the rumen without any change in the chemical structure [54]. By increasing the diet content of C18:2 n6c and C18:3 n3, for example by feeding hay or, better, fresh forages [56,57], also the milk concentration of this two fatty acids increases [58]. The other component of PC2, the C15:0, is mainly synthesized by rumen microflora but it could also derive from mammary gland de novo synthesis and directly from diet fatty acids. The increase of odd chain fatty acids in milk has been proved to be related to the increase of forage:concentrate ratio, because of an increase in cellulolytic bacteria, responsible for their synthesis [59]. That effect has also been demonstrated by different studies, both on milk and cheese [60,61]. Falchero et al. [62] have also linked the presence of high concentration of C15:0 and C17:0 in dairy products to the cow consumption of forages with elevated concentrations of these fatty acids. Thus, the PC2 was composed of three fatty acids that are strictly linked to the forage:concentrate ratio.

Finally, C21:0, C20:5 n3, and CLA n9c, n11t formed PC3. C21:0 is a long chain saturated fatty acid, which presence in milk depends on the ruminal microbial biohydrogenation of the forage fatty acids [63]. C20:5 n3 (eicosapentaenoic acid (EPA)) is a nutritionally important n3 PUFA which constitutes a really low portion of total fatty acids in milk [64]. It has been shown to be difficult to increase EPA concentration in milk by increasing its presence in the ruminants diet [65]. On the other hand, feeding cattle with high C18:3 n3 content forages (in particular, fresh grass [57]) seems to be an effective strategy to raise the EPA content in dairy products [66]. That effect is due to the elongation and desaturation of C18:3 n3 that occurs in ruminants tissues [58]. CLA n9c, n11t originates both from C18:2 n6c biohydrogenation in rumen and from endogenous Δ9-desaturation of vaccenic acid (C18:1 n11t), which in turn derives from C18:2 n6c and C18:3 n3 biohydrogenation [67]. It has been demonstrated that higher concentration of C18:3 n3 in the cattle diet leads to an higher concentration of CLA n9c, n11t in dairy products [67]. Thus, the high presence of C18:3 n3 in the ruminal environment, achieved by feeding hay or especially fresh forages [52], is responsible for the increasing concentration of both EPA and CLA n9c, n11t in milk, even if we also have always to consider that other factors can modify this relation [68]. Therefore, the fresh forages-based diet is responsible for the PC3 fatty acids association.

### 4.2. Characterization of Fatty Acids Profile on Milk and Cheese from Grazing and No Grazing Dairy Farming Systems

Both for milk and cheese, the PC2 included the fatty acids profile that best discriminated between the two farming systems, followed by PC3 and, lastly, PC1. These findings indicated that, between all the fatty acids that have been analysed, C18:2 n6c, C18:3 n3, and C15:0 are the ones that most make it possible to classify a product as belonging to grazing or no grazing system. On the other hand, the profile of saturated and monounsaturated fatty acids with 18 carbon atoms was influenced by the farming system to a lesser extent.

The importance of the PC2 and PC3 in the characterization of the two farming systems could be due, firstly, to the different intake of C18:3 n3 and C18:2 n6c between the animals of the two farms. Indeed, these fatty acids are mostly found in the fresh herbage to which the animals of the grazing system had access. The process of collection and drying of fresh forages for the production of hay leads to a loss of PUFA, in particular C18:3 n3, due to both oxidative reactions and to the leaves fall, which contain more fatty acids than stems [30]. The C18:2 n6c concentration in hay is also reduced by the process of wilting [69]. Thus, the lower intake of PUFA of cows fed hay, compared to animals fed fresh forages, will produce a lower presence of C18:3 n3 and C18:2 n6c in the rumen, a lower production of their biohydrogenation reaction intermediates (such as CLA n9c, n11t, and vaccenic acid), and a lower synthesis of their deriving metabolites (such as again CLA n9c, n11t, C20:5 n3, and other n3 PUFA). So, the concentration of all these fatty acids will be reduced in milk, compared with cows fed fresh forages. Regarding C15:0, its variable content in milk could have two different explanations: it could be due to the greater consumption through grazing of vegetable species which contain it in high concentration, or, most probably, it could be due to the higher forage:concentrate ratio of the grazing cows’ diet, which influence the ruminal fermentations and leads to a higher bacterial synthesis of this fatty acid.

Therefore, the function extracted with the DA, including the content of C18:3 n3, C18:2 n6c, and C15:0 as coefficients, could be a reliable tool for classifying dairy products. Hence, the dairy industries could reassure consumers about milk and cheese origin. Thus, an appropriate label and a standardized grazing-raise-dairy-product recognition system would be very beneficial, so the consumer could be encouraged to purchase milk and cheese from grazed cows, also considering that northern Italian consumers recognise the relation between type of feed and dairy products quality and tend to prefer milk coming from grazed cows [70]. Finally, that would mean that farmers who practice pasture system would be encouraged by a higher economic return.

The fatty acids members of PC1 (C14:0, C14:1 n9, C16:0, C18:0, and C18:1 n9c) did not contribute to the discrimination of the farming system. That is probably due to the fact that C14:0, C14:1 n9, and part of C16:0 are strictly dependent on the de novo synthesis in mammary gland, which occurs from volatile fatty acids (VFAs). The remaining part of C16:0 originates from fatty acids synthetized by ruminant tissues or directly derived from the feed [52], and that are not influenced by the hay making process. The same applies to the C18:0 and C18:1 n9c because of their presence in milk is independent from the unsaturation degree of the C18 fatty acids in the forage. In fact, it is always stearic acid which gives origin to the C18:0 and half of the C18:1 n9c in milk [71]. Thus, the desaturation degree of C18 fatty acids feed with the diet does not matter.

These results are not in agreement with others previous studies which identified C16:0 as the main fatty acid which could help in the classification of fresh grass feeding vs. other feeding system [32,72]. On the other hand, Gaspardo et al. [73] have also recognised an important role to unsaturated fatty acids and to fatty acids longer than 18C in reflecting in milk fatty acids profiles the different composition of rations (in particular with reference to forage:concentrate ratio). This is also confirmed by the study of Capuano et al. [32] that, in addition to C16:0, has recognised an important role to two different odd chain fatty acids and to several C18 fatty acids.

After cross-validation, the 83.3% and 87.5% of milk and cheese samples were correctly classified. The six grazing system’s misclassified samples were entirely, except one, due to the winter diet which is identical between the two farming systems and that, therefore, made it difficult to distinguish the different samples. This confirms, as expected, that the diet during the grazing significantly differentiates the characteristics of milk and cheese fatty acids profile. The misclassification of the cheese sample of November represents an unexpected result, which does not have a definitive explanation. On the other hand, just one sample of the no grazing farm was misclassified and was the one collected on the 1st October, thus, immediately after the supplementation of fresh grass. It has been shown that the first significant changes in fatty acids profile, after that the cows are taken off pasture, occur in a few days, but the stabilization of the concentration of the different fatty acids needs between 4 (CLA n9c, n11t) and 7 days (C18:3 n3 and C18:2 n6c) [74]. So, this could explain the misclassification.

Thus, the milk and cheese samples that were mostly misclassified have been the ones collected during no grazing season. This result is in agreement with the study of Capuano et al. [32] that have found difficulties in the discrimination of samples collected in spring. As elucidated by them, for our model this is a “good mistake”. In fact from a practical point of view, we are not interested in knowing if a particular dairy product derives from a farm which practice grazing but in this moment keeps its cows in barn; our goal was to identify products that derive from cows that were grazing on the day in which milking was carried out.

Moreover, the distribution of the scores in the factor map evidences the higher variability of fatty acids profiles of milk and cheese of the grazing system compared to the no grazing one. This is due to the diet major variability of the grazing system cows which change every six months. In addition, during grazing season, cows have the possibility to choose between all the different vegetable species present on the pasture, that make their diet more differentiated than the one of the no grazing system. In fact, the ration of the latter is more constant in its components and this explicate also the easiness of the identification of its samples, being that it reflects in a milk and cheese’s more constant fatty acids composition along the year.

Although the use of two farms that only differed in the grazing reduced the sources of confounders, further research for the validation of our discriminant function is required, including a larger number of dairy farms and a wider period of sampling. In addition, a further study could evaluate the effect of the grazing area.

## 5. Conclusions

Our study firstly has described the fatty acids composition of milk and cheese of Original Brown, an Italian native breed whose role, by a socio-economic point of view, is constantly growing in different North Italy provinces. Secondly, we have underlined the relationships existing between the most relevant fatty acids in dairy products linked to the rumen fermentation processes and de novo synthesis of fatty acids in mammary gland, and lastly, to the diet composition. Finally, we managed to define a model which could help in discriminate between grazing and no grazing system dairy products, starting from the concentration of a small number of fatty acids in milk or cheese. This model, after validation, could be an important tool to the dairy industry. Indeed, the present study proposes a new methodological approach to the characterization of milk and cheese based on the acidic profile and multivariate analysis although it requires an external validation.

## Figures and Tables

**Figure 1 animals-10-01231-f001:**
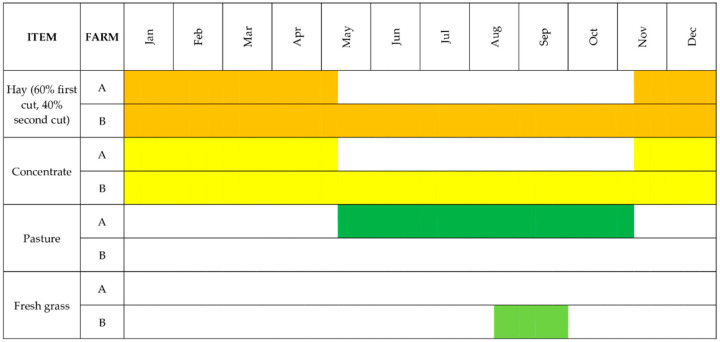
Rations composition of the two different farming system (“A” farm practiced grazing, “B” farm not) during the experimental period.

**Figure 2 animals-10-01231-f002:**
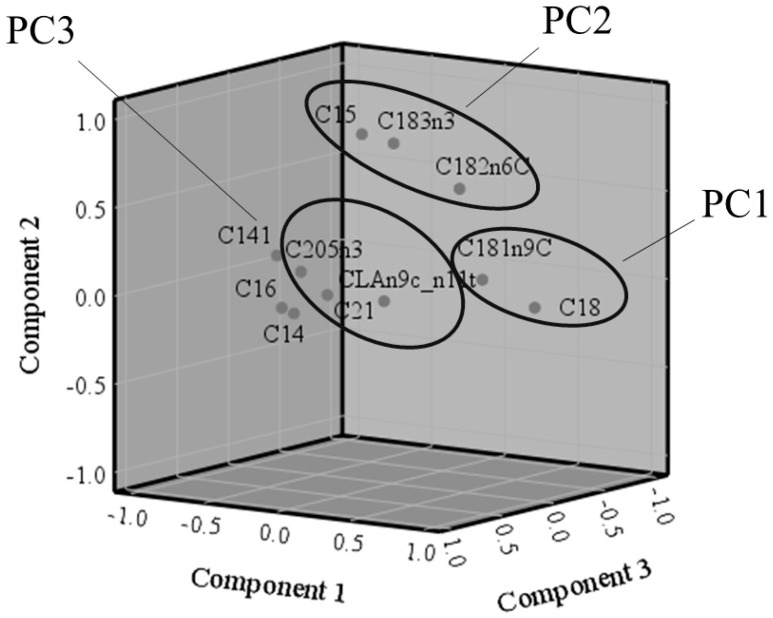
Component plot in rotated space. The circles indicate variables with positive loading >0.8 for each principal component.

**Figure 3 animals-10-01231-f003:**
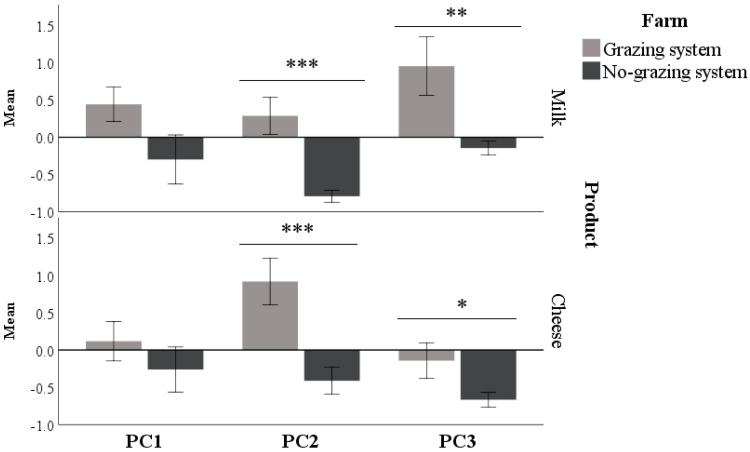
Mean ± standard error of the component (PC) scores in milk (top panel) and cheese (lower panel) of grazing and no grazing farms. “PC1. C < 18 − C18” had positive loadings for saturated and monounsaturated fatty acids with 18 carbon atoms; “PC2. C18 PUFA & C15” included PUFA n6 and n3 with 18 carbon atoms in addition to pentadecanoic acid. PC3. C > 18 & CLA had positive loadings for fatty acids containing 20 or more carbon atoms and for the conjugated linoleic acid.

**Figure 4 animals-10-01231-f004:**
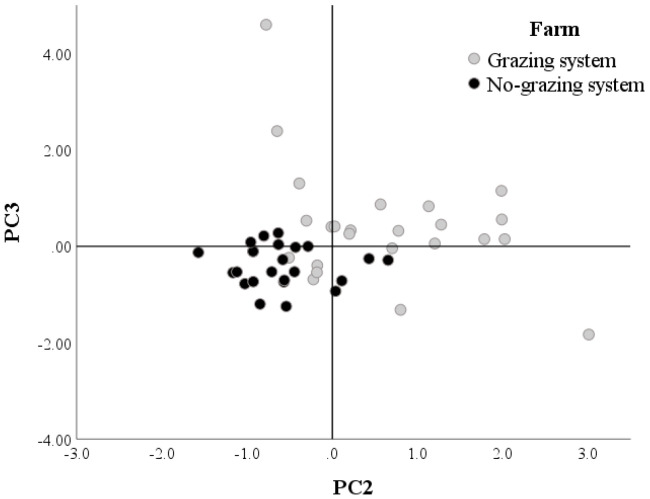
Factor maps of the principal component analysis (PCA). Distributions of the scores concerning the PC2. C18 PUFA & C15 (*x*-axis) and PC3. C > 18 & CLA (*y*-axis) extracted after PCA. The grey circles show that the scores of the grazing system are concentrated on the first quadrant in which both coordinates are positive. This suggests that many samples of grazing farm had high content of PUFA, pentadecanoic acid, C > 18, and CLA. On the contrary, many points of the no-grazing farm (black circle) have negative coordinates (3rd quadrant) suggesting a lower content PUFA, pentadecanoic acid, C > 18, and CLA. It can also be noted that the circles of the grazing system are more scattered than the ones of the no grazing one. This suggests that the products of the grazing farm showed greater variability in their acid profile. The diet of the grazing system cows varies across the seasons and is responsible both for a major variability of its dairy products fatty acids profiles and for a major concentration of PC2 and PC3 fatty acids. On the other hand, in the no grazing system the ration is kept constant in its composition during all year, which means a minor variability in the fatty acid profiles, and, caused by the nearly complete absence of fresh forages, a minor concentration, compared to the grazing system, of PC2 and PC3 fatty acids.

**Table 1 animals-10-01231-t001:** Mean and standard deviation (SD) of detected fatty acids in milk from A (grazing system) and B (no-grazing system) farms.

Fatty Acids	Farm				Tot	
	A		B			
	Mean	SD	Mean	SD	Mean	SD
**Saturated fatty acids**						
C6:0	0.63	0.20	0.62	0.47	0.63	0.01
C8:0	0.62	0.28	0.42	0.38	0.52	0.14
C10:0	2.01	0.41	2.20	0.40	2.10	0.14
C11:0	0.08	0.08	0.10	0.07	0.09	0.02
C12:0	2.74	0.41	3.34	0.42	3.04	0.42
C13:0	0.06	0.02	0.05	0.02	0.06	0.01
C14:0	11.52	1.12	13.37	1.41	12.44	1.31
C15:0	1.80	0.29	1.52	0.10	1.66	0.20
C16:0	32.17	3.36	36.79	4.80	34.48	3.27
C17:0	0.94	0.22	0.80	0.18	0.87	0.10
C18:0	11.75	1.81	10.28	2.40	11.01	1.04
C20:0	0.22	0.06	0.19	0.03	0.20	0.03
C21:0	0.07	0.06	0.03	0.01	0.05	0.03
C22:0	0.11	0.03	0.08	0.01	0.09	0.02
C23:0	0.05	0.02	0.04	0.01	0.05	0.01
C24:0	0.09	0.02	0.06	0.01	0.07	0.02
***Tot SFA***	64.85	4.04	69.89	3.94	67.37	3.57
**Monounsaturated fatty acids**						
C14:1 n9	0.95	0.17	1.10	0.25	1.02	0.11
C16:1 n9	1.53	0.42	1.57	0.40	1.55	0.03
C18:1 n6c	1.01	1.59	1.73	2.72	1.37	0.51
C18:1 n6t	0.30	0.17	0.37	0.13	0.33	0.05
C18:1 n9c	23.17	2.25	19.53	3.33	21.35	2.57
C18:1 n9t	2.97	1.80	2.31	2.31	2.64	0.47
C20:1 n9	0.18	0.05	0.10	0.05	0.14	0.06
C22:1 n9	0.02	0.01	0.01	0.01	0.02	0.01
C24:1 n9	0.02	0.01	0.01	0.00	0.02	0.01
***Tot MFA***	30.14	3.23	26.72	3.36	28.43	2.41
**n6 Polynsaturated fatty acids**						
C18:2 n6c	1.76	0.41	1.55	0.30	1.66	0.15
C18:2 n6t	0.28	0.24	0.08	0.05	0.18	0.14
C18:3 n6	0.01	0.00	0.01	0.00	0.01	0.01
C20:2 n6	0.04	0.02	0.03	0.01	0.04	0.01
C20:3 n6	0.06	0.01	0.06	0.01	0.06	0.00
C20:4 n6	0.09	0.01	0.10	0.01	0.09	0.00
C22:2 n6	0.05	0.03	0.03	0.01	0.04	0.01
***Tot n 6 PUFA***	2.29	0.37	1.85	0.31	2.07	0.31
**n3 Polynsaturated fatty acids**						
C18:3 n3	1.44	0.25	0.75	0.11	1.09	0.48
C20:3 n3	0.02	0.01	0.02	0.01	0.02	0.01
C20:5 n3	0.09	0.03	0.06	0.01	0.07	0.02
C22:6 n3	0.01	0.00	0.01	0.00	0.01	0.00
***Tot n3 PUFA***	1.56	0.25	0.84	0.12	1.20	0.51
**Stereoisomers of conjugated linoleic acid**						
CLA n9c, n11t	1.41	0.70	0.96	0.37	1.18	0.32
**n6/n3 ratio**	1.47	0.28	2.22	0.56	1.73	0.53

The concentrations are expressed in g (100 g)^−1^ of milk. SFA = saturated fatty acids; MUFA = monounsaturated fatty acids; PUFA = polyunsaturated fatty acids; CLA = conjugated linoleic acids; n6 = omega-6 fatty acids; n3 = omega-3 fatty acids; c = cis; and t = trans.

**Table 2 animals-10-01231-t002:** Mean and SD of detected fatty acids in cheese from A (grazing system) and B (no-grazing system) farms.

Fatty Acids	Farm				Tot	
	A		B			
	Mean	SD	Mean	SD	Mean	SD
**Saturated fatty acids**						
C6:0	0.36	0.31	0.29	0.28	0.33	0.05
C8:0	0.65	0.20	0.49	0.36	0.57	0.12
C10:0	2.24	0.37	2.32	0.65	2.28	0.05
C11:0	0.06	0.08	0.15	0.10	0.11	0.06
C12:0	2.99	0.43	3.53	0.52	3.26	0.38
C13:0	0.05	0.05	0.05	0.03	0.05	0.01
C14:0	12.32	1.19	13.92	1.34	13.12	1.13
C15:0	1.89	0.31	1.61	0.18	1.75	0.20
C16:0	33.16	2.83	35.55	3.56	34.35	1.69
C17:0	1.05	0.15	0.80	0.17	0.92	0.18
C18:0	11.17	1.91	10.54	2.43	10.85	0.45
C20:0	0.21	0.06	0.18	0.02	0.20	0.03
C21:0	0.05	0.02	0.03	0.01	0.04	0.01
C22:0	0.12	0.04	0.08	0.01	0.10	0.03
C23:0	0.04	0.03	0.05	0.01	0.05	0.01
C24:0	0.07	0.02	0.06	0.01	0.06	0.01
***Tot SFA***	66.44	3.16	69.62	2.99	68.03	2.25
**Monounsaturated fatty acids**						
C14:1 n9	1.03	0.14	1.10	0.25	1.07	0.06
C16:1 n9	1.52	0.50	1.45	0.48	1.49	0.05
C18:1 n6c	1.89	2.91	1.44	2.08	1.67	0.32
C18:1 n6t	0.32	0.13	0.37	0.15	0.34	0.036
C18:1 n9c	21.75	2.85	20.44	3.29	21.10	0.92
C18:1 n9t	2.53	1.62	2.38	0.82	2.45	0.11
C20:1 n9	0.10	0.08	0.10	0.05	0.10	0.01
C22:1 n9	0.02	0.02	0.01	0.00	0.02	0.01
C24:1 n9	0.03	0.03	0.01	0.00	0.02	0.01
***Tot MFA***	29.18	2.62	27.31	2.86	28.24	1.33
**n6 Polynsaturated fatty acids**						
C18:2 n6c	1.94	0.32	1.61	0.24	1.78	0.23
C18:2 n6t	0.13	0.07	0.07	0.04	0.10	0.04
C18:3 n6	0.02	0.01	0.01	0.00	0.01	0.01
C20:2 n6	0.06	0.08	0.04	0.01	0.05	0.02
C20:3 n6	0.07	0.05	0.06	0.02	0.06	0.01
C20:4 n6	0.07	0.04	0.09	0.02	0.08	0.02
C22:2 n6	0.03	0.02	0.03	0.01	0.03	0.01
***Tot n 6 PUFA***	2.32	0.37	1.91	0.26	2.12	0.29
**n3 Polynsaturated fatty acids**						
C18:3 n3	1.48	0.38	0.81	0.14	1.15	0.47
C20:3 n3	0.02	0.00	0.02	0.00	0.02	0.01
C20:5 n3	0.07	0.03	0.05	0.01	0.06	0.01
C22:6 n3	0.05	0.07	0.01	0.00	0.03	0.03
***Tot n3 PUFA***	1.62	0.37	0.89	0.16	1.26	0.52
**Stereoisomers of conjugated linoleic acid**						
CLA n9c, n11t	0.73	0.37	0.42	0.09	0.58	0.22
**n6/n3 ratio**	1.43	0.27	2.15	0.54	1.68	0.51

The concentrations are expressed in g (100 g)^−1^ of cheese. SFA = saturated fatty acids; MUFA = monounsaturated fatty acids; PUFA = polyunsaturated fatty acids; CLA = conjugated linoleic acids; n6 = omega-6 fatty acids; n3 = omega-3 fatty acids; c = cis; and t = trans.

**Table 3 animals-10-01231-t003:** Loadings of factors extracted with the principal component analysis.

Item	Component		
	*PC1.* *C < 18 − C18*	*PC2.* *C18 PUFA & C15*	*PC3.* *C > 18 & CLA*
C18:0	**0.947**	−0.025	−0.047
C16:0	**−0.893**	−0.235	−0.208
C14:0	**−0.861**	−0.274	−0.281
C14:1 n9	**−0.853**	0.081	−0.103
C18:1 n9c	**0.725**	0.139	0.147
C18:3 n3	0.222	**0.882**	0.294
C15:0	−0.123	**0.873**	0.110
C18:2 n6c	0.474	**0.609**	0.011
C21:0	0.174	0.101	**0.870**
C20:5 n3	−0.062	0.197	**0.791**
CLA n9c, n11t	0.423	0.062	**0.674**
**% Variance explained**	45.7	18.8	11.9
**Cumulative % variance explained**	76.4		

Factor loadings with an absolute value greater than 0.5 are in bold.

**Table 4 animals-10-01231-t004:** Standardized canonical discriminant function coefficients and Wilks’ lambda of variables included in the discriminant analysis to discriminate the fatty acids profile in milk and cheese from grazing and no grazing system.

Product	Variable	Standardized Canonical Discriminant Function Coefficients	Wilks’ Lambda	*p* Value
**Milk**	***PC2.*** ***C18 PUFA & C15***	0.901	0.567	<0.001
	***PC3.*** ***C > 18 & CLA***	0.766	0.747	0.012
	***PC1. C < 18 − C18***	0.396	0.867	0.079
**Cheese**	***PC2.*** ***C18 PUFA & C15***	0.988	0.618	0.001
	***PC3.*** ***C > 18 & CLA***	0.780	0.843	0.055
	***PC1. C < 18 − C18***	0.766	0.961	0.355

The variables are sorted in decreasing order of importance for each product.

**Table 5 animals-10-01231-t005:** Classification results of milk and cheese samples after cross validation. The table crosses the original classification (in the rows) with that obtained from the solution of the discriminant analysis (in the columns) to examine the validity of the solution. Values are number and percentage.

Product	Farm	Predicted Group Membership		Total
		Grazing System	No Grazing System	
**Milk ^1^**	**Grazing system**	9 (75.0)	3 (25.0)	12
	**No grazing system**	1 (8.3)	11 (91.7)	12
**Cheese ^2^**	**Grazing system**	9 (75.0)	3 (25.0)	12
	**No grazing system**	0 (0.0)	12 (100.0)	12

^1^ 83.3% of cross-validated grouped cases correctly classified.^2^ 87.5% of cross-validated grouped cases correctly classified.

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
