# Peer review of "Determination of Fatty Acids Profile in Original Brown Cows Dairy Products and Relationship with Alpine Pasture Farming System"

_animals, 2020, doi:10.3390/ani10071231_

Round 1

Reviewer 1 Report

The authors have answerd my questions and made a note in the manuscript on the validation of the model/tool.

I have no further comments and suggest to accept the manuscript.

Author Response

Dear editor,

We truly believe that all the suggestions made by the reviewers have helped in improving our manuscript. We have made the last modifications as suggested by Reviewer 2.

REVIEWER 1

Comment: The authors have answered my questions and made a note in the manuscript on the validation of the model/tool.

I have no further comments and suggest to accept the manuscript.

Response: we do really thanks the Reviewer for the given advices and for the appreciation about the modifications we have carried out.

Reviewer 2 Report

I have read the revised version of the manuscript entitled: "Determination of fatty acids profile in Original Brown cows dairy products and relationship with alpine pasture farming system". The authors answered all my requests and the quality of presentation is improved. I only suggest two minor revisions:

  • Figure 2: Report, in the capture, the complete fatty acids name for a better comprehension (e.i. CLA911= CLA n9c, n11t).
  • Supplementary tables: check names throughout the manuscript. Tables should be called Table S1, S2, S3, without the M.

Author Response

REVIEWER 2

Comment: I have read the revised version of the manuscript entitled: "Determination of fatty acids profile in Original Brown cows dairy products and relationship with alpine pasture farming system". The authors answered all my requests and the quality of presentation is improved. I only suggest two minor revisions:

Figure 2: Report, in the capture, the complete fatty acids name for a better comprehension (e.i. CLA911= CLA n9c, n11t).

Response: we are very pleased that the Reviewer has appreciated our modifications of the manuscript. In Figure 2 we have changed the word “CLA911” with “CLAn9c_n11t”. The reason why the new denomination doesn’t still completely match with the correct nomenclature, is that the program utilized for the production of this type of graphic doesn’t allow the use of  special characters. Thus, that is the best solution we can achieve.

Comment: Supplementary tables: check names throughout the manuscript. Tables should be called Table S1, S2, S3, without the M.

Response: we thank the Reviewer for having noticed these inconsistences. We have checked all the manuscript and corrected any mistakes.

Reviewer 3 Report

This Reviewer had no major changes required.  The authors appear to have addressed the comments from Reviewer #2.

Author Response

REVIEWER 3

Comment: This Reviewer had no major changes required.  The authors appear to have addressed the comments from Reviewer #2.

Response: we thanks again the Reviewer for the given suggestions.

This manuscript is a resubmission of an earlier submission. The following is a list of the peer review reports and author responses from that submission.

Round 1

Reviewer 1 Report

This study evaluated a potential relationship between fatty acid patterns in milk and cheese based on the use of alpine summer-grazing.  The species of cattle evaluated was the Original Brown cow.  The am was to determine if fatty acid profiling can distinguish dairy products of Original Brown cows grazing or not on Italian alpine pasture diets.  Milk and cheese were sampled every month.

In a detailed methodologic description of statistics, Discriminant analysis (DA) was performed in milk and cheese products to identify fatty acid combinations that distinguish grazing and no grazing.

Results showed that milk and cheese from no grazing farming system had lower C-18 PUFA, C-15, C>18 and CLA than grazing system.  The two feeding methods had unique fatty acids profile of milk and cheese.  The authors discuss that these differences may be due to different intake of C18 fatty acids as these fatty acids are mostly found in the fresh herbage.

The article as whole is a sound study.  Results are complex, but presented as well as they can be.  Conclusions are reasonable.

Author Response

Comment: This study evaluated a potential relationship between fatty acid patterns in milk and cheese based on the use of alpine summer-grazing.  The species of cattle evaluated was the Original Brown cow.  The aim was to determine if fatty acid profiling can distinguish dairy products of Original Brown cows grazing or not on Italian alpine pasture diets.  Milk and cheese were sampled every month.

In a detailed methodologic description of statistics, Discriminant analysis (DA) was performed in milk and cheese products to identify fatty acid combinations that distinguish grazing and no grazing.

Results showed that milk and cheese from no grazing farming system had lower C-18 PUFA, C-15, C>18 and CLA than grazing system.  The two feeding methods had unique fatty acids profile of milk and cheese.  The authors discuss that these differences may be due to different intake of C18 fatty acids as these fatty acids are mostly found in the fresh herbage.

The article as whole is a sound study.  Results are complex, but presented as well as they can be.  Conclusions are reasonable.

Response: we thank the Reviewer for this careful analysis. We are very pleased that the Reviewer appreciated our approach which aimed to simplify the understanding of such complex and not always intuitive methodologies. As suggested, we have checked for English and typographical errors.

Reviewer 2 Report

I have read the manuscript entitled "Relationship and determination of fatty acids profiles to identify alpine pasture-raised dairy products from Original Brown cow". The aim of the paper is very interesting and overall the manuscript is well written. Some english errors/typos are present, please carefully check and edit. The statistical approach is appropriate. Nonetheless, some improvement are necessary concerning the presentation of data and the general editing: 

  • The Title does not totally reflect the aim of the work and is not clear. I suggest rephrasing it to something more direct (e.g. Determination of fatty acids profile in Original Brown cows dairy products and relationship with alpine pasture farming system.)
  • Line 175-176 this sentence does not belong to the M&M section but to results. Accordingly Table 1: (that should also be edited) should be move to results and not to be mentioned in line 189.
  • Table 1: I would suggest to present the descriptive variables divided for the two farms.  
  • The results section is somehow chaotic and hard to read. It should be more cohesive and straightforward to allow for a better comprehension. Make sure that the fatty acids nomenclature is always the some (e.g. table 1 vs table 2). 
  • Author contributions: remove the template explanation (Conceptualization should be the first word, line 470)
  • Acknowledgments: remove this section if there is none. 
  • Check references (e.g. line 168 the three references are mixed up)

Author Response

Comment: I have read the manuscript entitled "Relationship and determination of fatty acids profiles to identify alpine pasture-raised dairy products from Original Brown cow". The aim of the paper is very interesting and overall the manuscript is well written. Some english errors/typos are present, please carefully check and edit. The statistical approach is appropriate.

Response: we thank the Reviewer for his suggestions. We have followed them and believe that they have contributed to improve our manuscript. We thank the Reviewer, in particular, for the suggested title, which we consider very appropriate.

Comment: Nonetheless, some improvement are necessary concerning the presentation of data and the general editing: the Title does not totally reflect the aim of the work and is not clear. I suggest rephrasing it to something more direct (e.g. Determination of fatty acids profile in Original Brown cows dairy products and relationship with alpine pasture farming system.)

Response: we very thank the Reviewer for this valuable suggestion. We have changed the title with the one recommended.

Comment: Line 175-176 this sentence does not belong to the M&M section but to results. Accordingly Table 1: (that should also be edited) should be move to results and not to be mentioned in line 189.

Response: we agree with the Reviewer and moved the fatty acids composition to the Results section.

Comment: Table 1: I would suggest to present the descriptive variables divided for the two farms. 

Response: as suggested, we divided according to the farming system.

Comment: The results section is somehow chaotic and hard to read. It should be more cohesive and straightforward to allow for a better comprehension. Make sure that the fatty acids nomenclature is always the some (e.g. table 1 vs table 2).

Response: we understand the Reviewer that the presentation of results is not intuitive because complex analyzes have been performed and a little familiarity with the multivariate approach is necessary. We tried to simplify by referring only to the number of PCs and separating the different paragraphs. Moreover, we have added two tables (Table 3S and Table 4) to better present the results of the discriminating analysis. Finally, after the rough results, a hint of interpretation was presented.

We have checked the nomenclature and now it’s consistent throughout all the manuscript.

Comments: Author contributions: remove the template explanation (Conceptualization should be the first word, line 470).

Acknowledgments: remove this section if there is none.

Response: as suggested, we have removed the indicated sentences.

Comment: Check references (e.g. line 168 the three references are mixed up).

Response: as suggested we have checked all the references and corrected the mistakes.

Reviewer 3 Report

Dear authors,

i have read your manuscript and believe it is interesting. I have a number of comments:

Major comment:

1)you are using only 2 herds of which you take a monthly bulk milk sample. Each herd represents a production system.

That means that you have only 6 samples representing bulk milk of which the cows are grazing (herd A: April-September).

In my opinion this is a limited number to build a prediction model on to identify if milk/cheese is produced while cows are grazing. I know you are using loo, but these cows are still coming from the same 2 herds.

I would like to see if the prediction would also work if you would test bulk milk samples of independent herds from a different grazing area.

Minor comments:

line 94: even if... should be even though

line 106: ...an hay-based... should be ... a hay based ...

line 120: ... to distinguishing... should be ... to disctinghuish ...

line 134: ... was not the grazing practice. should be ...was not due to the grazing practice.

lines 131-145: what is the distibution of parity and days in milk of the cows in both farms?

Line 162: how old were the cheeses when you samples these? you say 30 days would be optimal (line 153), did you match the optimal time-point?

Author Response

Comment: Dear authors,

i have read your manuscript and believe it is interesting. I have a number of comments:

Major comment:

1)you are using only 2 herds of which you take a monthly bulk milk sample. Each herd represents a production system. That means that you have only 6 samples representing bulk milk of which the cows are grazing (herd A: April-September).

In my opinion this is a limited number to build a prediction model on to identify if milk/cheese is produced while cows are grazing. I know you are using loo, but these cows are still coming from the same 2 herds.

I would like to see if the prediction would also work if you would test bulk milk samples of independent herds from a different grazing area.

Response:

We understand the perplexity of the Reviewer about the 6 months of grazing. In this regard, we admit that we had thought to further divide the samples according to the pasture (and therefore to the months of production). However, the following reasons have dissuaded us: 1) we would have further complicated the analyzes that are already not intuitive to understand, 2) our goal was to propose an approach to distinguish the two managements in general, so that the grazing system farm could be characterized in its annual production, 3) the effect of grazing is presumably prolonged when the cows return to the tie-stall barn and it is difficult to predict and speculate the duration of this “pasture effect” in milk and, above all, in cheese.

We also agree with the other observations of the Reviewer. For this reason, we stressed the limit of the sample size and highlighted in the Conclusions that this work proposes a new methodological approach and it is not a model that can already be applied (L. 493-495, 503-506). The need for validation is also presented in the Abstract (L 57-58). We hope we have thus stressed the need for external validation suggested by the Reviewer.

Finally, we used the observation on the possible effect of the grazing area to insert them as future prospects (L. 494-495).

Comment: Minor comments: line 94: even if... should be even though

Response: revised as requested.

Comment: line 106: ...an hay-based... should be ... a hay based ...

Response: revised as requested.

Comment: line 120: ... to distinguishing... should be ... to disctinghuish ...

Response: revised as requested.

Comment: line 134: ... was not the grazing practice. should be ...was not due to the grazing practice.

Response: revised as requested.

Comment: lines 131-145: what is the distribution of parity and days in milk of the cows in both farms?

Response: as suggested, we added the information about average days in milk and distribution of parity.

Comment: Line 162: how old were the cheeses when you samples these? you say 30 days would be optimal (line 153), did you match the optimal time-point?

Response: we thank the Reviewer for the question. We have clarified this point in the manuscript.

Reviewer 4 Report

This study is very interesting and the experimental design and statistical evaluation has been carried out to a high standard.

I have made some minor suggestions in the attached file, but otherwise believe that it is acceptable for publication. 

Author Response

Comment: This study is very interesting and the experimental design and statistical evaluation has been carried out to a high standard.

Response: we thank the Reviewer for these encouraging words and for the right suggestions. We have followed them all and below are the details.

Comment: I have made some minor suggestions in the attached file, but otherwise believe that it is acceptable for publication.

line 30: replace “vitally” with “vital”

Response: revised as requested.

Comment: line 39: de novo should be in italics throughout the manuscript.

Response: revised as requested.

Comment: line 40: replace “define” with “defining”.

Response: revised as requested.

Comment: line 46: eliminate “2018’s”.

Response: revised as requested.

Comment: line 48: add “in the 2018 season”.

Response: revised as requested.

Comment: line 62: eliminate “up”

Response: revised as requested.

Comment: line 80: eliminate “until”

Response: revised as requested.

Comment: line 94: add “it”

Response: revised as requested.

Comment: line 95: replace “performs” with “performance”

Response: revised as requested.

Comment: line 97: add “a”

Response: revised as requested.

Comment: line 98: replace “impossibility” with “difficulty”

Response: revised as requested.

Comment: line 106 : replace “condition” with “systems”

Response: revised as requested.

Comment: line 110: replace “relation” with “relationship”

Response: revised as requested.

Comment: line 113 : replace “the consumer” with “consumers”

Response: revised as requested.

Comment: line 120: eliminate “yet”

Response: revised as requested.

Comment: line 125: add “Analysis”

Response: revised as requested.

Comment: line 375: replace “that …reach” with “achieved”

Response: revised as requested.

Comment: line 400 : replace “explication” with “explanation”

Response: revised as requested.

Comment: line 406 : Please clarify this, the current sentence structure is unclear.

Response: we thank the reviewer for this suggestion. We have edited the sentence in order to clarify the concept.

Comment: line 407 : eliminate “thanks” and add “would be very beneficial, so”

Response: revised as requested.

Comment: line 415: ?

Response: we modified this sentence.

Comment: line 417: replace “goes for the” with “applies to”

Response: revised as requested.

Comment: line 423 : See also O'Callaghan et al 2016, Journal of Dairy Science 99: 9424-9440

Response: we do really thank the Reviewer for this suggestion, we have read the article and it has been included in the References.

Comment: line 433 : replace “manage to give an” with “have a definitive”

Response: revised as requested.